# Predictors and Long-Term Outcomes of Pathological Complete Response Following Neoadjuvant Treatment and Radical Surgery for Locally Advanced Rectal Cancer

**DOI:** 10.3390/jcm14124251

**Published:** 2025-06-15

**Authors:** Dan Assaf, Yaacov Lawrence, Ofer Margalit, Einat Shacham-Shmueli, Lior Bear, Nadav Elbaz, Alexander Lebedayev, Edward Ram, Yasmin Anderson, Ofir Gruper, Michael Goldenshluger, Lior Segev

**Affiliations:** 1Division of General Surgery, Sheba Medical Center, Ramat Gan 5262000, Israel; 2Department of Oncology, Sheba Medical Center, Ramat Gan 5262000, Israel; 3Faculty of Medicine, Tel-Aviv University, Tel-Aviv 6997801, Israel

**Keywords:** rectal cancer, neoadjuvant therapy, pathology, outcomes

## Abstract

**Background:** Pathological complete response (pCR) following neoadjuvant therapy and surgery for locally advanced rectal cancer is associated with improved prognosis. Accurately predicting who will achieve pCR could theoretically eliminate the need for surgery for these patients. We aimed to compare pCR and non-pCR rectal cancer patients following neoadjuvant therapy, searching for clinical predictors for pCR and comparing oncological outcomes between these groups. **Methods**: This is a single-center retrospective analysis of all patients who underwent a curative-intent rectal resection between 2010 and 2020 for primary non-metastatic rectal cancer following neoadjuvant therapy. The cohort (263 patients) was divided into two groups according to the pathological results from surgery: the pCR group (53 patients) and the non-pCR group (210 patients). **Results**: The groups were similar in terms of baseline characteristics, clinical presentation, and staging, but tumors of the pCR group were significantly higher in the rectum (mean distance from the anal verge 7.92 cm versus 6.9 cm respectively, *p* = 0.04), and more of them were located at the posterior rectal wall (37.7% versus 24.3%, *p* = 0.049). Multivariate analysis found posterior location and tumor height to be significantly associated with pCR (OR 2.23, 95% CI 1.11–4.45, *p* = 0.023), (OR 1.14, 95% CI 1.03–1.27, *p* = 0.015). The 5-year overall survival was 95.6% in the pCR group compared with 87.5% in the non-pCR group (*p* = 0.09), and the 5-year disease-free survival was 92.7% versus 64.5%, respectively (*p* < 0.001). **Conclusions**: Tumor location at the posterior wall of the rectum and higher tumor location were found to be associated with pCR. Patients achieving pCR demonstrate improved prognosis compared with non-pCR patients.

## 1. Introduction

Approximately one third of rectal cancer patients are diagnosed with locally advanced disease at diagnosis [1], which is defined as tumor invasion beyond muscularis propria or into regional lymph nodes. The conventional treatment for locally advanced rectal cancer (LARC) involves neoadjuvant long-course chemoradiotherapy (LCCRT) or short-course radiotherapy (SCRT) followed by total mesorectal excision (TME) surgery and adjuvant chemotherapy. This sandwich regimen promotes tumor downstaging, increases resectability, enhances sphincter preservation [2], reduces local recurrence rates [3], and leads to pathological complete response (pCR) in 10–30% of patients [4,5]. However, this approach has limited impact on distant metastases, the leading cause of treatment failure, and adherence to adjuvant chemotherapy is less than 50% due to treatment complications and poor compliance [6]. As a result, total neoadjuvant therapy (TNT) has gained popularity in recent years, in which systemic chemotherapy and LCCRT are administered prior to surgery. This approach aims to reduce rates of distant metastasis, decrease treatment-related toxicity, improve compliance, and improve the pCR rate [6]. pCR is defined as the complete absence of viable tumor cells on pathological examination after surgery. It has been shown that pCR plays a significant role in predicting the prognosis of patients treated with neoadjuvant regimens, and those who achieve pCR have a significantly better prognosis than those who do not achieve pCR [7]. In patients with a pCR, the 2-year disease-free survival and overall survival have been reported to reach up to 98 and 100%, respectively [8]. Patients with a clinical diagnosis of complete response (CCR) may opt for a non-operative management of the “watch-and-wait” approach, which avoids surgery and its associated morbidity (38.8% overall complication rate and 11.8% grade ≥ 3 complications [9]) and therefore improve patients’ quality of life [10,11]; however, accurate pathological staging would not be available. Accurately predicting which patients will achieve pCR still remains a significant challenge. Although the disappearance of gross tumor following neoadjuvant therapy has been found to be highly correlated with pCR, residual disease within the mesorectal lymph nodes has been reported in up to 17% of patients [12]. Therefore, identifying clinical markers predictive of pCR is essential. Various factors have been suggested in the past to predict pCR, including preoperative carcinoembryonic antigen (CEA) levels, circumferential extent of the tumor, and tumor distance from the anal verge [13].

The purpose of this study was to compare pCR and non-pCR rectal cancer patients following neoadjuvant therapy in order to find clinical predictors for pCR and to compare the oncological outcomes of the two groups.

## 2. Materials and Methods

**Design/population** The study was conducted in single center and approved by the institutional review board. We retrospectively reviewed all consecutive curative-intent radical rectal resection between the years 2010 and 2020 identified with relevant surgery procedure codes including abdomino-perineal resection (APR). Only patients diagnosed with primary clinical stage II or III rectal adenocarcinoma who underwent neoadjuvant therapy prior to surgery were included in the final analysis. The cohort was divided into two groups according to the pathological results from surgery: the pCR group and the non-pCR group.

**Neoadjuvant therapy options** The decision between the different neoadjuvant protocols was made during a multidisciplinary team meetings considering all patient and disease parameters and included the following: short course radiotherapy (SCRT) which consisted of 1 week of radiation in 5 fractions (median radiation dose of 25 Gy), long course chemoradiotherapy (LCCRT) usually including 5 weeks of radiation in 25 fractions (median radiation dose of 50.4 Gy) together with 5-FU-based chemotherapy, and total neoadjuvant therapy (TNT) which included either LCCRT followed by a course of systemic chemotherapy (FOLFOX or FOLFIRI or XELOX) or systemic chemotherapy followed by LCCRT. Immunotherapy was not a part of neoadjuvant protocols.

**Variables and outcome measures** We reported patient characteristics and demographics, preoperative disease characteristics including tumor location in the rectum (lower rectum < 5 cm, mid rectum 5–10 cm, upper rectum > 10 cm) and neoadjuvant oncological therapy, operative characteristics, postoperative surgical outcomes including 30-days postoperative morbidity graded according to the Clavien–Dindo classification system [14]. Clavien–Dindo grade > 2 was defined as a major complication. Histopathological surgical results included tumor size, differentiation, pathological stage, number of harvested lymph nodes, and distance of tumor from distal margins. Long-term oncological outcomes included details about adjuvant radiation therapy or adjuvant chemotherapy, recurrence of local or distant disease, and mortality data. Disease-free survival (DFS) was defined as the period from the date of surgery to the date of the first recurrence. If tumor recurrence was not recorded, DFS was defined as the time between the date of surgery and the last follow-up date. Overall survival (OS) was calculated from the date of surgery to either the date of death or the last follow-up visit.

**Statistical analysis** Data were gathered into a REDCap database (Vanderbilt University, Nashville, Tenn, USA). Data were analyzed using SPSS version 29 software (Armonk, NY, USA: IBM Corp). All statistical analyses were conducted with a significance level of α = 0.05.

Descriptive statistics are presented using prevalence and percentage values for categorical variables, means and standard deviation for continuous variables, and medians and ranges for non-continuous variables.

Groups were compared using Student’s t-test for continuous normally distributed variables and the Mann–Whitney test for non-parametric comparison. Categorical comparisons used the χ^2^ test and Fisher’s exact tests as required.

Univariate analysis for association with pathological response was conducted by binary logistic regression, and multivariate analysis for association with pathological response was performed using a binary logistic regression model with forward conditional method to exclude non-significant variables from the model.

Survival analysis was conducted using a Kaplan–Meier curve and the log rank test for significance. Multivariate survival analysis was performed using a Cox regression test with the forward conditional method to exclude non-significant variables from the model.

## 3. Results

### 3.1. Patients’ Characteristics

This study included 263 patients, of which 53 patients were in the pCR group and 210 patients in the non-pCR group, corresponding to a pCR rate of 20%. The median age was 62.3 (range 31–88), and the groups’ demographics patterns were similar (Table 1).

### 3.2. Clinical Presentation

The two groups were similar in terms of clinical presentation and staging at diagnosis; however, tumors in the pCR group were located higher in the rectum (mean distance from the anal verge was 7.92 cm versus 6.9 cm, respectively, *p* = 0.04), and more of them were located at the posterior wall of the rectum (37.7% versus 24.3%, respectively, *p* = 0.049) (Table 2).

### 3.3. Perioperative Characteristics

The operative and perioperative characteristics were also similar between the groups, including intra-operative and postoperative morbidity (Table 3). However, more patients in the pCR group had their stoma reversed (79.2% versus 60% among the non-pCR group, *p* = 0.009). There were significantly higher rates of laparoscopic approach among the pCR group (58.5% versus 37.1%, *p* = 0.005). Although fewer patients in the pCR group underwent robotic surgery (20.8% versus 29.5%), this difference was not statistically significant (*p* = 0.2). Consequently, there was no significant difference in the overall use of minimally invasive approaches between the two groups (79.2% among the pCR group versus 66.7% among the non-pCR, *p* = 0.07) (Table 3).

The pathological report of the non-pCR group is detailed in Table 4. The mean tumor size in pathological report was 2.4 cm, and the median number of harvested lymph nodes was 13 (range 0–53) among the non-pCR patients compared with only 11 nodes (2–38) among the pCR patients (*p* = 0.043). Only three patients (1.4%) had positive radial margins. No patient had positive distal resection margins.

A significantly lower proportion of patients in the pCR group received adjuvant chemotherapy compared with the non-pCR group (43.4% versus 64.8%, *p* = 0.004) (Table 3).

### 3.4. Predictors of Pathological Complete Response

A logistic regression model searching for variables associated with pCR found tumor location in the posterior rectal wall to be significantly associated with pCR in univariate (OR 1.89, 95%CI 1–3.58, *p* = 0.05) and multivariate (OR 2.23, 95% CI 1.11–4.45, *p* = 0.023) analysis. Tumor distance from the anal verge (higher tumors) was also found to be significantly associated with pCR in univariate (OR 1.11, 95% CI 1–1.22, *p* = 0.05) and multivariate analysis (OR 1.14, 95% CI 1.03–1.27, *p* = 0.015) (Table 5). Other variables, including age, gender, smoking habits, clinical staging, CEA levels, and type of neoadjuvant treatment were not found to be associated with pCR.

### 3.5. Long-Term Oncological Outcomes

The median follow-up time was 60.7 months (range 4–179) with no differences in follow-up times between the groups. During this follow-up period, only 3 patients among the pCR group (5.7%) experienced disease recurrence compared with 67 patients among the non-pCR group (31.9%, *p* < 0.001). There were no local recurrences among the pCR group (Table 3). The 5-year overall survival rate was 95.6% in the pCR group compared with 87.5% in the non-pCR group (*p* = 0.09). The 5-year disease-free-survival rate was 92.7% in the pCR group compared with 64.5% in the non-pCR group (*p* < 0.001) (Figure 1).

Cox regression analysis confirmed that pCR was significantly associated with improved DFS in univariate (HR = 0.152, 95% CI 0.05–0.48, *p* = 0.001) and multivariate analysis (HR = 0.176, 95% CI 0.06–0.57, *p* = 0.004) (Table 6).

Higher post-treatment CEA levels (HR = 1.013, 95% CI 1.003–1.02, *p* = 0.015) and a greater number of positive lymph nodes (HR = 1.12, 95% CI 1.05–1.19, *p* < 0.001) were also associated with worse DFS in both univariate and multivariate analysis. Conversion from a minimally invasive approach to an open procedure was also found to be associated with decreased DFS in multivariate analysis (HR = 0.47, 95% CI 0.24–0.91, *p* = 0.026); however, this was not significant in univariate analysis (Table 6).

## 4. Discussion

In this study, the rate of pCR following neoadjuvant therapy were 20%. This is in line with the previously published literature where the pCR rates following LCCRT vary between 10 and 30% [4,5,15]. TNT protocols were only introduced toward the end of our study period. As a result, only a small subset of patients in our cohort received TNT with the vast majority still undergoing LCCRT. We deliberately chose not to expand our study period beyond 2020 so that we could evaluate long-term oncological outcomes. TNT protocols are expected to raise pCR rates [6,7].

We found tumor location to be strongly associated with pCR—specifically, posterior tumors and tumors located higher in the rectum. Reports in previous studies regarding the association of tumor height and pCR are not consistent. In concordance with our results, Runau et al. found that tumors further from the anal verge were more likely to develop pCR, and this was confirmed in multivariate analysis that showed an increased odds ratio of 1.046 for achieving pCR for every centimeter above the anal verge [16]. This may support the hypothesis that accurate and effective radiotherapy delivery is more challenging for lower tumors, thereby reducing the likelihood of pCR. Similarly, a study of 332 patients with middle and low rectal cancer identified middle tumor level as a predictor of pCR compared with low rectal tumors [17]. There is a debate as to whether upper rectal cancer (>10 cm) should undergo neoadjuvant therapy before surgery, as those tumors show lower local recurrence rates compared with middle and low rectal tumors [18]. Conversely, Das et al. [13] reported in a series of 526 patients that greater tumor distance from the anal verge independently predicted lower rates of tumor downstaging in multivariate analysis (*p* = 0.01).

Our observation that posterior tumor location is associated with pCR is novel. This may be reasonable compared with circumferential tumors which are expected to have a poorer response to neoadjuvant treatment; however, it is not clear why posterior tumors had a better response than anterior tumors. There is no difference in the application of radiation between posterior and anterior tumors. We speculate whether anterior tumors have higher rates of T4 tumors (due to the relative proximity to other organs), which might be less responsive to radiotherapy. Previous studies have not addressed the association between different rectal wall tumor locations and response to neoadjuvant therapy. These findings warrants further research.

CEA is well established as a prognostic marker in colorectal cancer for diagnosis, follow-up, and monitoring response to therapy [19]. Although in this study we did not find a direct correlation between CEA levels and pCR, we found that higher levels of post-treatment CEA were associated with worse DFS. Multiple studies have reported low CEA as a significant factor associated with pCR [17,20,21,22,23]. There is, however, still some debate whether pre-treatment CEA, post-treatment CEA, or both are more predictive of pCR.

Interestingly, in our study, there was no correlation between pre-treatment clinical tumor staging and pCR. Specifically clinical nodal status was not found to be associated with pCR as could have been expected, although it was associated with disease recurrence. On the other hand, a study by Shin et al. analyzing 1089 consecutive rectal cancer patients who underwent radical resection after chemoradiotherapy revealed that clinical N-positive stage, tumor size > 4 cm, and poorly differentiated tumors were significantly associated with lower odds of pCR [24]. Zhou et al. [25] found that patients with longitudinal tumor length <5 cm were more likely to achieve pCR (24% versus 9.4%, *p* = 0.027), although it was not significant in multivariate analysis. They also found negative extramural vascular invasion (EMVI) to be an independent predictor of pCR in univariate (*p* = 0.014) and multivariate analysis (*p* = 0.045). Xu et al. [23] reported a cut-off value of 5.75 cm tumor size above which it was less likely to achieve pCR; however, tumor size was not found to be an independent predictor in multivariate analysis. Despite inconsistent finding across studies, it seems reasonable that pre-treatment clinical staging factors such as nodal status, tumor size, and other related features affect the response to neoadjuvant therapy. Accordingly, we believe that further attention and research should be devoted to this topic.

The number of harvested lymph nodes in the pCR group was significantly lower in our study compared with the non-pCR patients. In a similar manner, Runau et al. demonstrated that a lower lymph node yield significantly correlates with obtaining pCR (mean lymph nodes examined 7.48 versus 9.94, *p* = 0.004) [16]. However, this was not significant in multivariate analysis. A cohort study from Cleveland Clinic [26] found that a lymph node yield < 12 had more pCR (36% versus 19%, *p* = 0.01), and none of the <12-nodes patients experienced local recurrence. In addition, Beppu et al. found that the response of the primary tumor to chemoradiotherapy correlates strongly with the response of positive lymph nodes and that the requirements for total regression of positive lymph nodes are degeneration of the primary tumor and lymph node regression < 6 mm [27]. These findings have clinical implication for radiological reassessment for post-neoadjuvant patients when considering clinical complete response, patients with few lymph nodes (<8 combined with size < 6 mm) and no mucosal lesion may be considered for a watch-and-wait protocol.

A more contemporary systematic review [28] assessing predictors for pCR during the current TNT era has grouped the predictors into four categories: biochemical predictors, clinical predictors, patient demographics, and the order of therapy sequence. Biochemical predictors that enhance tumor susceptibility to TNT include normal pretreatment CEA levels and absence of genetic mutations (wild type); however, the current quality of data for biochemical and genetic markers of pCR after TNT is low. Clinical negative predictors for pCR included sarcopenia, hypoalbuminemia, positive circumferential resection margin, clinical stage N2 and T4, tumor diameter > 5 cm, and CEA > 5 after treatment. However, the authors report that the quality of evidence for clinical predictors of pCR following TNT is low as well. Assessing the response of patients to TNT by age had widely varying results due to small sample sizes which decreased the power of the studies. As for the order of therapy sequence, consolidation chemotherapy appears to be superior to induction chemotherapy as a predictor for pCR [28].

Consistent with the previous literature (10,15), our study demonstrated markedly better long-term oncological outcomes among the pCR patients compared with the non-pCR patients. The 5-year DFS was 92.7% in the pCR group compared with 64.5% in the non-pCR group (*p* < 0.001), and the 5-year OS was 95.6% in the pCR group compared with 87.5% in the non-pCR group (*p* = 0.09). Xu et al. reported a 5-year DFS rate of 94.7% among the pCR patients versus 59.7% in the non-pCR, *p* = 0.002, and 5-year OS rates of 95.8% versus 80.1%, respectively, *p* = 0.019 [23]. Notably, none of the pCR patients in our study experienced local recurrence during the study period. Similarly, other studies have also observed that if a recurrence does occur, distant metastasis is the major pattern of recurrence in patients achieving pCR after neoadjuvant chemoradiotherapy [24,26]. Shin et al. observed that the few pCR patients who experienced recurrence typically had clinical T3-T4 tumors, node positivity, and tumor distance from AV < 5 cm. They suggested administering adjuvant chemotherapy for patients with pCR who presented with these features following chemoradiotherapy [24].

Our study has some limitations. First, its retrospective single-center design limits the ability to generalize our results. In addition, the analysis spanned a 10-year period. As such, there were probably some pathological reporting variability which may have caused some bias. Still, all specimens were reviewed by fully trained specialized pathologists in accordance with national standards. During the second half of this decade period, TNT protocols were introduced and applied. This has gradually changed neoadjuvant therapy in rectal cancer from the usual chemoradiation alone to higher rates of patients treated with TNT. This transition period started at the end of this study; therefore, only a few patients in the study were treated with TNT. This led to some heterogeneity in neoadjuvant protocols, which may have introduced bias into our results. Finally, our cohort represents a relatively small sample size, which may limit the statistical power of our findings. As such, the credibility and generalizability of the results may be affected by variations in sample size, and larger studies are needed to confirm these observations.

## 5. Conclusions

In conclusion, our study has shown that a significant proportion (20%) of patients with locally advanced rectal cancer developed pCR following neoadjuvant therapy. These patients subsequently benefited from a better prognosis compared with the non-pCR patients. We identified posterior wall tumor location and higher tumor location as factors associated with pCR. Prospective contemporary studies that would incorporate current TNT and immunotherapy neoadjuvant protocols are needed to better define predictive factors of pCR in modern rectal cancer management. Future studies should aim at validating clinical factors, biochemical factors, and optimal TNT regimen to achieve pCR.

## Figures and Tables

**Figure 1 jcm-14-04251-f001:**
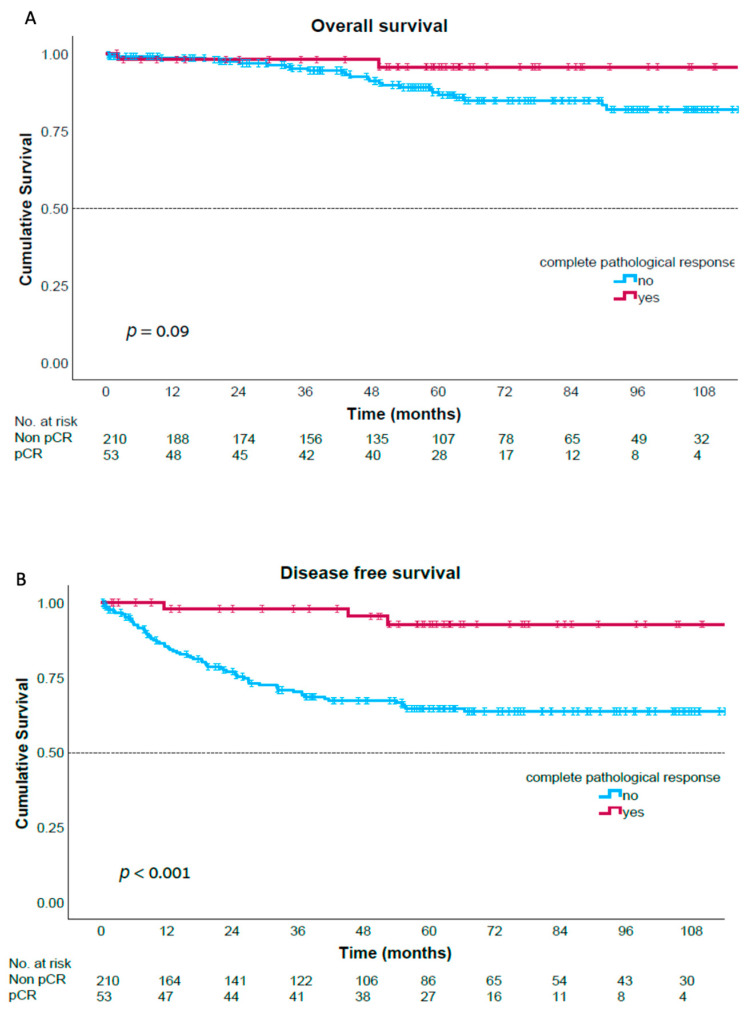
Overall survival (**A**) and disease-free survival (**B**) curves.

**Table 1 jcm-14-04251-t001:** Patient demographics and characteristics.

Variable	All Cohort (*n* = 263)	pCR Group (*n* = 53)	Non-pCR Group (*n* = 210)	*p* Value
Age in years, mean (range)	62.35 (31–88)	63.55 (34–82)	62.05 (31–88)	0.27
Gender				0.81
Male, *n* (%)	155 (58.9%)	32 (60.4%)	123 (58.6%)	
Female, *n* (%)	108 (41.1%)	21 (39.6%)	87 (41.4%)	
BMI, mean (range)	26.41 (15.6–41.1)	26.44 (15.6–35.4)	26.40 (16.4–41.1)	0.62
Past smoking, *n* (%)	57 (21.7%)	8 (15.1%)	49 (23.3%)	0.19
Current smoking, *n* (%)	58 (22.1%)	16 (30.2%)	42 (20%)	0.11
ASA score				
Median (range)	3 (1–5)	3 (1–3)	3 (1–5)	
1, *n* (%)	4 (1.8%)	1 (2.1%)	3 (1.7%)	0.81
2, *n* (%)	71 (32.1%)	14 (29.8%)	57 (32.8%)	0.92
3, *n* (%)	138 (62.4%)	32 (68.1%)	106 (60.9%)	0.24
4, *n* (%)	7 (3.2%)	0	7 (4%)	0.35
Comorbidities prevalence				
CVA/TIA, *n* (%)	7 (2.7%)	0	7 (3.3%)	0.35
Asthma/COPD, *n* (%)	24 (9.1%)	6 (11.3%)	18 (8.6%)	0.59
IHD, *n* (%)	32 (12.2%)	5 (9.4%)	27 (12.9%)	0.49
Arrythmia, *n* (%)	11 (4.2%)	2 (3.8%)	9 (4.3%)	0.99
DM, *n* (%)	66 (25.1%)	16 (30.2%)	50 (23.8%)	0.34
CRF, *n* (%)	7 (2.7%)	1 (1.9%)	6 (2.9%)	0.99
HTN, *n* (%)	106 (40.3%)	18 (34%)	88 (41.9%)	0.29
Dyslipidemia, *n* (%)	91 (34.6%)	19 (35.8%)	72 (34.3%)	0.83
Family hx CRC				
1st degree, *n* (%)	45 (17.1%)	8 (15.1%)	37 (17.6%)	0.66
Non 1st degree, *n* (%)	21 (8%)	3 (5.7%)	18 (8.6%)	0.58
Surgical hx, *n* (%)	108 (41.1%)	21 (39.6%)	87 (41.4%)	0.81
Bowel surgery hx, *n* (%)	27 (10.3%)	4 (7.5%)	23 (11%)	0.47

BMI—body mass index, ASA—American Society of Anesthesiologists, DM—diabetes mellitus, HTN—hypertension, IHD—ischemic heart disease, CRF—chronic renal failure, CVA—cerebral-vascular-accident, TIA—transient ischemic attack, COPD—chronic obstructive pulmonary disease, CRC—colorectal cancer.

**Table 2 jcm-14-04251-t002:** Clinical presentation.

Variable	All Cohort (*n* = 263)	pCR Group (*n* = 53)	Non-pCR Group (*n* = 210)	*p* Value
Symptoms present, *n* (%)	244 (92.8%)	46 (86.8%)	198 (94.3%)	0.07
Abdominal pain, *n* (%)	57 (21.7%)	8 (15.1%)	49 (23.3%)	0.19
Anemia, *n* (%)	21 (8%)	3 (5.7%)	18 (8.6%)	0.58
Weight loss, *n* (%)	74 (28.1%)	10 (18.9%)	64 (30.5%)	0.09
BM change, *n* (%)	138 (52.5%)	27 (50.9%)	111 (52.9%)	0.80
LGIB, *n* (%)	179 (68.1%)	32 (60.4%)	147 (70%)	0.18
Tumor location				
Lower rectum, (<5 cm), *n* (%)	53 (20.2%)	7 (13.2%)	46 (21.9%)	0.16
Mid rectum, (5–10 cm), *n* (%)	145 (55.1%)	31 (58.5%)	114 (54.3%)	0.58
Upper rectum, (>10 cm), *n* (%)	65 (24.7%)	15 (28.3%)	50 (23.8%)	0.49
Distance from AV, mean (range)	7.18 (0–15)	7.92 (2–15)	6.99 (0–15)	**0.04**
Clinical stage				
Stage 1, *n* (%)	7 (2.7%)	2 (3.8%)	5 (2.4%)	0.63
Stage 2, *n* (%)	60 (22.8%)	10 (18.9%)	50 (23.8%)	0.44
Stage 3, *n* (%)	196 (74.5%)	41 (77.4%)	155 (73.8%)	0.59
T stage				
T1, *n* (%)	4 (1.5%)	0	4 (1.9%)	0.59
T2, *n* (%)	14 (5.3%)	4 (7.5%)	10 (4.8%)	0.49
T3, *n* (%)	213 (81.0%)	43 (81.1%)	170 (81.0%)	0.98
T4, *n* (%)	32 (12.2%)	6 (11.3%)	26 (12.4%)	0.83
N stage				
N0, *n* (%)	67 (25.5%)	12 (22.6%)	55 (26.2%)	0.59
N1, *n* (%)	133 (50.6%)	31 (58.5%)	102 (48.6%)	0.19
N2, *n* (%)	63 (24%)	10 (18.9%)	53 (25.2%)	0.33
Rectal wall tumor location				
Anterior, *n* (%)	76 (28.9%)	12 (22.6%)	64 (30.5%)	0.26
Posterior, *n* (%)	71 (27%)	20 (37.7%)	51 (24.3%)	**0.049**
Circumferential, *n* (%)	40 (15.2%)	7 (13.2%)	33 (15.7%)	0.65
Neoadjuvant Tx:				
LCCRT *n*, (%)	208 (79.1%)	46 (86.8%)	162 (77.1%)	0.18
SCRT, *n* (%)	34 (12.9%)	4 (7.5%)	30 (14.3%)	0.11
TNT, *n* (%)	19 (7.2%)	3 (5.7%)	16 (7.6%)	0.77
CEA preoperative, mean (range)	6.82 (0–126.9)	4.09 (0–31.3)	7.54 (0–126.9)	0.29
CA 19-9, mean (range)	17.21 (0–248)	14.87 (0–92)	17.82 (0–248)	0.98
Preop albumin, mean (range)	4.03 (2.5–5.1)	4.1 (3–4.7)	4.02 (2.5–5.1)	0.25
Preop Hgb, mean (range)	12.65 (9–17.2)	12.9 (9.5–17.2)	12.59 (9–16.6)	0.19
Preop Creatinine, mean (range)	0.85 (0.2–2.49)	0.85 (0.2–1.54)	0.85 (0.45–2.49)	0.49

CEA—carcinoembryonic antigen, BM—bowel movements, LGIB—lower gastrointestinal bleeding, AV—anal verge, Hgb—hemoglobin, Tx—treatment, LCCRT—long course chemoradiotherapy, SCRT—short course radiotherapy, TNT—total neoadjuvant therapy.

**Table 3 jcm-14-04251-t003:** Operative and perioperative characteristics.

Variable	All Cohort (*n* = 263)	pCR Group (*n* = 53)	Non-pCR Group (*n* = 210)	*p*-Value
Surgical approach				
Open, *n* (%)	81 (30.8%)	11 (20.8%)	70 (33.3%)	0.07
Laparoscopic, *n* (%)	109 (41.4%)	31 (58.5%)	78 (37.1%)	**0.005**
Robotic, *n* (%)	73 (27.8%)	11 (20.8%)	62 (29.5%)	0.2
Conversion, *n* (%)	34 (12.9%)	7 (13.2%)	27 (12.9%)	0.95
Surgery type				
Anterior resection, *n* (%)	17 (6.5%)	3 (5.7%)	14 (6.7%)	0.99
Low anterior resection, *n* (%)	214 (81.4%)	46 (86.8%)	168 (80%)	0.26
Abdominoperineal resection, *n* (%)	31 (11.8%)	3 (5.7%)	28 (13.3%)	0.12
Additional procedure, *n* (%)	48 (18.3%)	8 (15.1%)	40 (19%)	0.51
Additional Anastomosis, *n* (%)	6 (2.3%)	0	6 (2.9%)	0.60
BSO, *n* (%)	23 (8.7%)	5 (9.4%)	18 (8.6%)	0.79
Hysterectomy, *n* (%)	6 (2.3%)	0	6 (2.9%)	0.60
Hernia, *n* (%)	4 (1.5%)	2 (3.8%)	2 (1%)	0.18
Cholecystectomy, *n* (%)	1 (0.4%)	0	1 (0.5%)	0.99
Intra-operative complications, *n* (%)	46 (17.5%)	5 (9.4%)	41 (19.5%)	0.08
Bleeding, *n* (%)	11 (4.2%)	1 (1.9%)	10 (4.8%)	0.70
Ureter injury, *n* (%)	4 (1.5%)	0	4 (1.9%)	0.59
Enterotomy, *n* (%)	14 (5.3%)	1 (1.9%)	13 (6.2%)	0.31
Splenic injury, *n* (%)	3 (1.1%)	0	3 (1.4%)	0.99
Vaginal injury, *n* (%)	6 (2.3%)	0	6 (2.9%)	0.60
Anastomotic disruption, *n* (%)	4 (1.5%)	2 (3.8%)	2 (1%)	0.18
Stoma diversion, *n* (%)	251 (95.4%)	50 (94.3%)	201 (95.7%)	0.67
Stoma reversal, *n* (%)	168 (63.9%)	42 (79.2%)	126 (60%)	**0.009**
LOS (days), median (range)	9 (4–73)	8 (4–47)	9 (5–73)	0.67
Postoperative complication, *n* (%)	189 (71.9%)	36 (67.9%)	153 (72.9%)	0.48
SSI, *n* (%)	34 (12.9%)	5 (9.4%)	29 (13.8%)	0.39
Abscess, *n* (%)	20 (7.6%)	4 (7.5%)	16 (7.6%)	0.99
Ileus/SBO, *n* (%)	60 (22.8%)	15 (28.3%)	45 (21.4%)	0.29
Anastomotic leak, *n* (%)	19 (7.2%)	5 (9.4%)	14 (6.7%)	0.55
Bleeding/transfusion, *n* (%)	29 (11%)	5 (9.4%)	24 (11.4%)	0.68
Pneumonia, *n* (%)	6 (2.3%)	1 (1.9%)	5 (2.4%)	0.99
UTI, *n* (%)	18 (6.8%)	2 (3.8%)	16 (7.6%)	0.54
DVT, *n* (%)	1 (0.4%)	0	1 (0.5%)	0.99
MI/Arrythmia, *n* (%)	9 (3.4%)	1 (1.9%)	8 (3.8%)	0.69
Wound dehiscence, *n* (%)	7 (2.7%)	0	7 (3.3%)	0.35
Electrolyte disturbances, *n* (%)	110 (41.8%)	20 (37.7%)	90 (42.9%)	0.49
Urinary retention, *n* (%)	23 (8.7%)	4 (7.5%)	19 (9%)	0.99
Clavien–Dindo score, median (range)	1 (0–5)	1 (0–3b)	1 (0–5)	0.92
Severe complications (CD score > 2), *n* (%)	49 (18.6%)	9 (16.9%)	40 (19%)	0.73
30-day readmission, *n* (%)	61 (23.2%)	13 (24.5%)	48 (22.9%)	0.79
30-day mortality, *n* (%)	2 (0.8%)	0	2 (1%)	0.99
Adjuvant chemotherapy, *n* (%)	159 (60.5%)	23 (43.4%)	136 (64.8%)	**0.004**
Recurrence, *n* (%)	70 (26.6%)	3 (5.7%)	67 (31.9%)	**<0.001**
Local recurrence, *n* (%)	18 (6.8%)	0	18 (8.6%)	**0.029**
Distant recurrence, *n* (%)	52 (19.7%)	3 (5.7%)	49 (23.3%)	**0.019**

BSO—bilateral salpingo-ophorectomy, LOS—length of stay, SSI—surgical site infection, SBO—small bowel obstruction, UTI—urinary tract infection, DVT—deep vein thrombosis, MI—myocardial infarction, CD—Clavien–Dindo.

**Table 4 jcm-14-04251-t004:** Subgroup non-pCR—pathology report.

Variable	Non-pCR Group (*n* = 210)
Pathological differentiation	
Poorly, *n* (%)	7 (3.3%)
moderately, *n* (%)	87 (41.4%)
well, *n* (%)	53 (25.2%)
missing, *n* (%)	63 (30.0%)
Pathological stage	
I, *n* (%)	74 (35.2%)
II, *n* (%)	65 (31.0%)
III, *n* (%)	69 (32.9%)
IV, *n* (%)	2 (1.0%)
T staging	
T0, *n* (%)	1 (0.5%)
T1, *n* (%)	15 (7.1%)
T2, *n* (%)	70 (33.3%)
T3, *n* (%)	118 (56.2%)
T4, *n* (%)	6 (2.9%)
N staging	
N0, *n* (%)	141 (66.2%)
N1, *n* (%)	55 (26.2%)
N2, *n* (%)	14 (6.7%)
Nx, *n* (%)	2 (1.0%)
Nodes examined, median (range)	13 (0–53)
Tumor size, cm, mean (range)	2.41 (0.1–10.2)
Positive nodes, median (range)	0 (0–16)
Metastasis, *n* (%)	2 (1.0%)
LVI, *n* (%)	14 (6.7%)
Perineural invasion, *n* (%)	24 (11.4%)
Distal margin, cm, median (range)	2 (0.2–7)
Radial margin involved, *n* (%)	3 (1.4%)
Signet ring cell, *n* (%)	8 (3.8%)
Mucinous tumor, *n* (%)	43 (20.5%)

LVI—lympho-vascular invasion, cm—centimeters.

**Table 5 jcm-14-04251-t005:** Logistic regression analysis for associations with pathological complete response.

	Univariate			Multivariate		
Variable	OR	95% CI	*p*	OR	95% CI	*p*
Age	1.01	0.99–1.04	0.42			0.45
Gender	0.93	0.5–1.72	0.81			0.73
Tumor Location						
Lower			0.36			0.64
Mid	0.51	0.19–1.36	0.18			0.96
upper	0.91	0.45–1.83	0.78			0.57
Distance from AV	1.11	1.00–1.22	**0.05**	1.14	1.03–1.27	**0.015**
Anterior rectal wall	0.67	0.33–1.35	0.26			0.68
Posterior rectal wall	1.89	1.00–3.58	**0.05**	2.23	1.1–14.45	**0.023**
Circumferential tumors	0.82	0.34–1.96	0.65			0.92
Clinical Staging						
Stage 1			0.66			0.73
Stage 2	1.51	0.28–8.08	0.63			0.65
Stage 3	0.75	0.35–1.62	0.47			0.48
Clinical T Stage						
T2			0.99			0.28
T3	1.73	0.4–7.47	0.46			0.42
T4	1.09	0.42–2.8	0.85			0.95
Clinical N Stage						
N1	1.16	0.46–2.90	0.76			0.62
N2	1.61	0.73–3.54	0.24			0.28
Long course chemoradiation			0.43			0.31
TNT	1.38	0.39–4.9	0.62			0.69
CEA preoperative	0.98	0.94–1.01	0.21			0.19
Past smoking	0.64	0.32–1.29	0.21			0.99
Current smoking	0.43	0.17–1.1	0.08			0.097

AV—anal verge, TNT—total neoadjuvant therapy, CEA—carcinoembryonic antigen.

**Table 6 jcm-14-04251-t006:** Cox regression analysis for associations with disease-free survival.

	Univariate			Multivariate		
Variable	HR	95% CI	*p*	HR	95% CI	*p*
pCR groups	0.152	0.05–0.48	**0.001**	0.176	0.06–0.57	**0.004**
Age	1.002	0.98–1.02	0.88			0.98
Gender	0.87	0.53–1.41	0.58			0.42
Distance from AV	0.93	0.86–1.01	0.09			0.21
Preoperative CEA (post-treatment)	1.014	1.003–1.03	**0.009**	1.013	1.003–1.02	**0.015**
Surgical approach						
Open	0.79	0.44–1.41	0.43			0.59
Minimal invasive	1.16	0.65–2.07	0.62			0.87
Conversion	0.73	0.38–1.38	0.33	0.47	0.24–0.91	**0.026**
Number of positive lymph nodes	1.12	1.05–1.19	**<0.001**	1.12	1.05–1.19	**<0.001**

pCR—pathological complete response, AV—anal verge, CEA—carcinoembryonic antigen.

## Data Availability

The data presented in this study are available upon request from the corresponding author due to privacy and ethical reasons.

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
