# Peer review of "Predictors and Long-Term Outcomes of Pathological Complete Response Following Neoadjuvant Treatment and Radical Surgery for Locally Advanced Rectal Cancer"

_jcm, 2025, doi:10.3390/jcm14124251_

Round 1
Reviewer 1 Report
Comments and Suggestions for Authors
Thank you for the opportunity to review the manuscript, “Predictors and long-term outcomes of pathological complete response following neoadjuvant treatment and radical surgery for locally advanced rectal cancer” by Assaf D. et al.
The authors investigated factors associated with pCR and the impact on long-term oncological outcomes in patients with locally advanced rectal cancer. They found that tumor location, specifically posterior and higher in the rectum, is associated with pCR, and patients with pCR tend to have better outcomes.
My comments and recommendations:
Introduction
Line 64 authors wrote: “...that avoid surgery and its substantial morbidity…” This statement is somewhat loaded. Surgery indeed carries an inherent risk of postoperative complications. However, in specialized colorectal surgery centers, the rates of postoperative complications are acceptable, as are the rates of curative oncological resection, maintaining a favorable risk-benefit ratio for rectal cancer patients. It is reasonable to assume that avoiding surgical resection provides the direct benefit of not having to deal with inconveniences, such as time away from work, postoperative recovery, costs, postoperative pain, scars, and long-term issues, such as bowel dysfunction. Nonetheless, this would not amount to substantial morbidity. It would be better if the authors replaced the statement with an objective report on the rates of postoperative complications to be avoided by non-operative management.
Statistical analysis
All statistical methods need to be reported here. The Cox regression needs to be reported as well.
Results
Line 163, the authors wrote: “the pCR group had their stoma reversed later on (79.2% versus 60% among the non-pCR group, p=0.009).” This is confusing. You are reporting a rate, not a time frame, so it makes no sense. What do you mean by later? What is the specific time frame?
Line 165/166 authors wrote: “but on the other hand, there were 165 less robotic procedures among them (20.8% versus 29.5% respectively, p=0.2).” This cannot be asserted, as the p-value of 0.2 is not significant.
3.5 Long-term oncological outcomes. The authors describe overall survival in the methods section and perform Kaplan-Meier analysis for overall survival. Therefore, they need to maintain consistency and report Cox regression for overall survival as well.
Lines 213-214. The Authors wrote: “The risk for disease recurrence is increased by 14% for each single lymph metastasis. For every 1 unit rise in CEA levels the risk for disease 214 recurrence is increased by 1.3%. A rise of 10 units in CEA is increasing the recurrence risk 215 by 13%.”
You cannot directly assume constancy just because you used the Cox regression. How was the constant hazard increase assessed? Is the proportional hazard assumption met? Alternatively, keep it simple and state that the hazard was 12% (not 14%) higher for an increase in positive lymph nodes. Same for CEA.
Discussion
Although well-organized and well-written, the discussion of the most interesting aspect of this study was dull and brief. There is evidence that lower rectal cancer has a higher incidence of local recurrence. Therefore, the finding that higher tumors have higher pCR rates, hence long-term outcomes, is appropriate. The discussion could expand this briefly with considerations for anatomy, tumor biology, and microenvironment. This would make for a more contemporary and engaging discussion for the readership.
Examples:
https://doi.org/10.3390/cancers15245853
https://pubmed.ncbi.nlm.nih.gov/29414633/
Author Response
|
Response to Reviewer 1 Comments
|
|||
|
1. Summary |
|
|
|
|
Thank you very much for taking the time to review this manuscript. Please find the detailed responses below and the corresponding revisions/corrections highlighted in the re-submitted files.
|
|||
|
2. Questions for General Evaluation |
Reviewer’s Evaluation |
Response and Revisions |
|
|
Does the introduction provide sufficient background and include all relevant references? |
Can be improved |
The introduction was revised according to the comments. |
|
|
Is the research design appropriate? |
Yes |
|
|
|
Are the methods adequately described? |
Can be improved |
The statistical methods section was improved according to the comments. |
|
|
Are the results clearly presented? |
Can be improved |
The results section was revised according to the comments below. |
|
|
Are the conclusions supported by the results? |
Can be improved |
The discussion was revised according to the comments. |
|
|
Are all figures and tables clear and well presented? |
Yes |
|
|
|
3. Point-by-point response to Comments and Suggestions for Authors |
|||
|
Comments 1: Introduction Line 64 authors wrote: “...that avoid surgery and its substantial morbidity…” This statement is somewhat loaded. Surgery indeed carries an inherent risk of postoperative complications. However, in specialized colorectal surgery centers, the rates of postoperative complications are acceptable, as are the rates of curative oncological resection, maintaining a favorable risk-benefit ratio for rectal cancer patients. It is reasonable to assume that avoiding surgical resection provides the direct benefit of not having to deal with inconveniences, such as time away from work, postoperative recovery, costs, postoperative pain, scars, and long-term issues, such as bowel dysfunction. Nonetheless, this would not amount to substantial morbidity. It would be better if the authors replaced the statement with an objective report on the rates of postoperative complications to be avoided by non-operative management.
|
|||
|
Response 1: Thank you for pointing this out. We have now replaced this statement and added an objective reference on the rates of postoperative complications. (https://doi.org/10.1111/codi.14376). In this international multicenter study the overall complication rate was 38.8%, and major complications (Clavien grade ³3) rate was 11.8%. This change can be found on page 2, first paragraph, line 64. |
|||
|
Comments 2: Statistical analysis All statistical methods need to be reported here. The Cox regression needs to be reported as well |
|||
|
Response 2: We appreciate the reviewer’s valuable comment. In response, we have revised the Methods section to include a comprehensive description of all statistical analyses performed in the study. Specifically, we added the Cox regression analysis. We hope this addresses the reviewer’s concern. See page 4, lines 123-137. |
|||
|
Comment 3: Results Line 163, the authors wrote: “the pCR group had their stoma reversed later on (79.2% versus 60% among the non-pCR group, p=0.009).” This is confusing. You are reporting a rate, not a time frame, so it makes no sense. What do you mean by later? What is the specific time frame? |
|||
|
Response 3: Thank you for pointing this out. We agree that the original phrasing was unclear. Our intention was to report the proportion of patients who eventually underwent stoma reversal, rather than the timing of the reversal. To clarify this, we have rephrased the sentence in line 163 to: "A greater proportion of patients in the pCR group had their stoma reversed compared to the non-pCR group." We have removed the word “later” to avoid any confusion regarding timing, as specific timeframes were not analyzed in this context. |
|||
|
Comment 4: Line 165/166 authors wrote: “but on the other hand, there were 165 less robotic procedures among them (20.8% versus 29.5% respectively, p=0.2).” This cannot be asserted, as the p-value of 0.2 is not significant. |
|||
|
Response 4: Thank you for your comment. We agree that the original phrasing may have implied a significant difference. We have revised the sentence to clarify that although the rate of robotic surgery was lower in the pCR group, this difference did not reach statistical significance (p=0.2). The revised sentence in line 165 now more accurately reflects the data and avoids any overinterpretation. See page 6, line 166. |
|||
|
Comment 5: 3.5 Long-term oncological outcomes. The authors describe overall survival in the methods section and perform Kaplan-Meier analysis for overall survival. Therefore, they need to maintain consistency and report Cox regression for overall survival as well. |
|||
|
Response 5: We appreciate the reviewer’s valuable comment. In response, we have revised the Methods section to include a comprehensive description of all statistical analyses performed in the study. Specifically, we added the Cox regression analysis. (page 11, line 212, and Table 6). We hope this addresses the reviewer’s concern. |
|||
|
Comment 6: Lines 213-214. The Authors wrote: “The risk for disease recurrence is increased by 14% for each single lymph metastasis. For every 1 unit rise in CEA levels the risk for disease 214 recurrence is increased by 1.3%. A rise of 10 units in CEA is increasing the recurrence risk 215 by 13%.” You cannot directly assume constancy just because you used the Cox regression. How was the constant hazard increase assessed? Is the proportional hazard assumption met? Alternatively, keep it simple and state that the hazard was 12% (not 14%) higher for an increase in positive lymph nodes. Same for CEA. |
|||
|
Response 6: Thank you for your insightful comment. We agree that the initial interpretation implied a constant hazard increase, which is not appropriate without formally assessing the proportional hazards assumption and the linearity of continuous predictors. We have revised the paragraph accordingly to directly report the hazard ratios without extrapolating percent increases. The revised text in line 215 now reflects a more accurate and cautious interpretation of the Cox regression findings. |
|||
|
Comment 7: Discussion Although well-organized and well-written, the discussion of the most interesting aspect of this study was dull and brief. There is evidence that lower rectal cancer has a higher incidence of local recurrence. Therefore, the finding that higher tumors have higher pCR rates, hence long-term outcomes, is appropriate. The discussion could expand this briefly with considerations for anatomy, tumor biology, and microenvironment. This would make for a more contemporary and engaging discussion for the readership. Examples: https://doi.org/10.3390/cancers15245853 https://pubmed.ncbi.nlm.nih.gov/29414633/
|
|||
|
Response 7: We appreciate this comment. We have expand our discussion accordingly regarding contemporary predictors for pCR referring to a recent systematic review which you suggested (https://doi.org/10.3390/cancers15245853). Key findings of predictors in this review included biochemical predictors such as genetic mutations, clinical predictors including sarcopenia, hypoalbuminemia, and clinical stage at diagnosis, and patient demographics including age, and the order of therapy sequence in TNT. However, the authors concluded that the overall quality of evidence was generally low, and currently there is not enough data to enable accurate prediction of pCR. See page 13, line 290. |
|||
|
4. Response to Comments on the Quality of English Language |
|||
|
Point 1: the English is fine and does not require any improvement |
|||
|
|||
Reviewer 2 Report
Comments and Suggestions for Authors
This study investigates the association between pathological complete response (pCR) following neoadjuvant therapy and surgery for locally advanced rectal cancer and improved prognosis. The primary objectives were to compare clinical characteristics and oncological outcomes between patients achieving pCR and those not achieving pCR (non-pCR) after neoadjuvant therapy, and to identify potential clinical predictors for pCR. Obviously, this is a clinically significant issue, particularly for patients who achieve pathological remission. For these individuals, the option of 'Watch and Wait' could be considered, thereby avoiding permanent harm caused by surgery. Therefore, it represents a topic of substantial clinical importance.
However, there are still several issues that require clarification from the authors:
1.The description of baseline characteristics in the article lacks important information such as tumor size and clinical tumor typing. This omission may have an impact on the results. Could the authors provide this missing baseline data?
2.In Table 4 (Subgroup non-pCR – pathology report), immunohistochemical results for special markers such as microsatellite instability (MSI) are not mentioned. Could the authors include these results or explain why they were not included?
3.The authors did not clearly state whether immunotherapy was used in the preoperative neoadjuvant treatment of the patients. This is also an aspect that could affect the results. Could the authors clarify this point?
4.The authors mentioned that patients with tumors located at the posterior rectal wall are more likely to achieve pCR. Could this be related to the easier setup of radiotherapy target areas at the posterior rectum (considering that there are fewer adjacent organs at the posterior wall, making it easier to achieve adequate radiotherapy)? Could the authors provide more clarification on this?
5.In the limitations section, the authors should explicitly state that the cases included in this study are of a small sample size, and the credibility of the results may change with variations in sample size. Could the authors include this statement?
6. The authors refer to "lower tumors" and "higher tumors" in relation to the distance from the anal verge (AV). To avoid confusion for readers, could the authors further clarify these terms?
7.As this study involves human subjects, information regarding ethics approval is required. Could the authors provide details about the ethics approval process for this study?
8.In the conclusions of the study, the authors should further define future research prospects based on the current results to guide future studies. Could the authors expand on this aspect?
Round 2
Reviewer 1 Report
Comments and Suggestions for Authors
Thank you for the opportunity to review the manuscript, “Predictors and long-term outcomes of pathological complete response following neoadjuvant treatment and radical surgery for locally advanced rectal cancer” by Assaf D. et al.
The authors investigated factors associated with pCR and the impact on long-term oncological outcomes in patients with locally advanced rectal cancer. They found that tumor location, specifically posterior and higher in the rectum, is associated with pCR, and patients with pCR tend to have better outcomes.
There were significant improvements in the revision. I recommend accepting. Thank you for the opportunity to participate in the revision of this manuscript.
Author Response
Comment: Thank you for the opportunity to review the manuscript, “Predictors and long-term outcomes of pathological complete response following neoadjuvant treatment and radical surgery for locally advanced rectal cancer” by Assaf D. et al.
The authors investigated factors associated with pCR and the impact on long-term oncological outcomes in patients with locally advanced rectal cancer. They found that tumor location, specifically posterior and higher in the rectum, is associated with pCR, and patients with pCR tend to have better outcomes.
There were significant improvements in the revision. I recommend accepting. Thank you for the opportunity to participate in the revision of this manuscript
Response: Thank you. We appreciate your review
Reviewer 2 Report
Comments and Suggestions for Authors
In Comment 1, I mentioned the classification of tumor gross specimens, such as ulcerative type, exophytic (protruding) type, and infiltrating type, as well as the statistical analysis of this baseline data. However, the author did not provide a clear response regarding this. Additionally, it appears that the newly introduced indicators were not included in the final univariate analysis, at least that's what is indicated.
Author Response
Comment: In Comment 1, I mentioned the classification of tumor gross specimens, such as ulcerative type, exophytic (protruding) type, and infiltrating type, as well as the statistical analysis of this baseline data. However, the author did not provide a clear response regarding this. Additionally, it appears that the newly introduced indicators were not included in the final univariate analysis, at least that's what is indicated.
Response: We appreciate your comment. Unfortunately, we do not have data regarding the gross specimen appearance besides the full pathological report as detailed in table 4. The newly introduced indicator (tumor size) is not included in the logistic regression univariate analysis because this indicator exists only for the non-pCR group, and the analysis has to include factors which are relevant to both groups. (the tumor size was taken from the operative pathological report, therefore, there is no tumor size for the pCR group).